# Influence of Milk Thistle (*Silybum marianum*) Seed Cakes on Biochemical Values of Equine Plasma Subjected to Physical Exertion

**DOI:** 10.3390/ani11010210

**Published:** 2021-01-16

**Authors:** Hana Dockalova, Ladislav Zeman, Pavel Horky

**Affiliations:** Department of Animal Nutrition and Forage Production, Mendel University in Brno, Zemědělská 1, 61300 Brno, Czech Republic; zeman@mendelu.cz (L.Z.); pavel.horky@mendelu.cz (P.H.)

**Keywords:** phytogenics, silymarin, horse nutrition, physical exercise, biochemical parameters

## Abstract

**Simple Summary:**

The use of milk thistle (*Silybum marianum*) is considered a safe phytogenic additive in animal nutrition. The aim of this study was to monitor the influence of feeding with milk thistle seed cakes included in the usual feed dose and to detect the effect on biochemical indicators of equine plasma. Milk thistle has been known for its positive effects on the liver (in liver diseases) and the antioxidant effects. Milk thistle seed cakes were fed as part of a normal feed dose in this study. Statistically significant differences were found out between the experimental and control groups of the horses. The significant values of the non-specific liver enzyme AST (aspartate transaminase) were determined in the experimental group. Statistically significant differences were detected after the exposure of the horses to physical exercise, especially for the values of cortisol, NEFA (non-esterified fatty acids), and inorganic phosphorus. Our results suggest that the feeding of milk thistle seed cakes can have a positive effect on the health of horses.

**Abstract:**

Veterinarians can recommend milk thistle for the treatment of equine liver disease and laminitis. Milk thistle seed cakes were fed in the range of normal feed doses in this study. The milk thistle seed cakes were fed (twice a day) to the experimental group of the horses (*n* = 5) and biochemical blood markers (TP, Albumin, ALT (alanine transaminase), AST (aspartate transaminase), ALP (alkaline phosphatase), GGT (gamma-glutamyltransferase), Bilirubin, Cholesterol, HDL (high-density lipoprotein), LDL (low-density lipoprotein), TAG (triacylglycerol), BHB (beta-hydroxybutyric acid), NEFA (non-esterified fatty acids), creatine kinase, creatinine, Urea, GSH-Px (glutathione peroxidase), TAS (total antioxidant status), lactate, glucose, cortisol, Ca, Pi) were monitored. The control group of horses (*n* = 5), bred and trained in the same conditions, was used for comparison. The control group received the entire feed dose as accepted by the horses in the experimental group before the beginning of the experiment. The aim was to find out whether the preparation of milk thistle seed cakes could have positive effects on the health of the horses. All ten horses received one feeding form before the beginning of the experimental monitoring. All horses were exposed to heavy physical exercise (regular combined driving training) after 56 days of milk thistle seed cakes feeding (up to 400 g/day). Three blood samples were taken (before physical exercise; about 15 min and 60 min after physical exercise). Significant differences (*p* < 0.05) were detected in the values of AST, NEFA, cortisol and Pi in the experimental group. The exercise effect was detected in the values of albumin, lactate, cortisol, NEFA, and calcium. Our results suggest that the feeding of milk thistle seed cakes could have a positive effect on the health of the horses.

## 1. Introduction

The use of phytogenic additives (namely *Silybum mariannum*) is ordinarily considered a safe supplement because herbal extracts bring therapeutic and nutritional benefits to horses. Phytogenic additives can prevent the development of stress-related health disorders within a normal feed dose [1]. A promising direction is the use of alternative feeds containing bioactive compounds or mixtures of natural origins or the use of phytoadditives or plant extracts, probiotics, prebiotics, symbiotics, or oilseed by-products in animal nutrition [2].

These safe phytogenic additives also include milk thistle [1]. Silymarin, a complex of active substances contained in milk thistle achene, has shown potential antioxidant, antimicrobial, anticancer, antidiabetic, cardiovascular-protective, hepatoprotective, and neuroprotective effects, among others. Silymarin has proven effects on counteracting the toxicities of antibiotics, metals, and pesticides [3]. Milk thistle seeds contain flavonoids, including quercetin, taxifolin, eriodyctiol, and chrysoeriol [4]. Up to 23 variants of flavonolignans have been isolated in *S. marianum* [5]. Silymarin is considered a dietary supplement recommended for dogs, cats, and horses with liver disease but a lack of clinical studies demonstrating the efficacy because the studies of these animals are limited [6]. Silymarin is also recommended for laminitis in horses. The endotoxins are neutralized, and lipopolysaccharides are reduced to induce lamellar separation, probably due to the antioxidant and anti-inflammatory effects of silymarin in vitro laminitis model [7]. Silymarin has also been reported to increase glutathione levels in the blood of horses [8]. Flavonolignans, coming from milk thistle, are hepatoprotectants also added to the animals as the xenobiotic protectives (the elimination of the effects of toxic substances such as pesticide residues, mycotoxins, etc.) [9].

It should also be noted that milk thistle seed cakes contain not only silymarin but can also be a source of other nutrients, especially the residual oil fraction. The oil fraction, containing linoleic, oleic, palmitic, sterols, tocopherol, and phospholipids, is neglected due to the focus on flavonolignans [10]. Further research is needed to elucidate the effects [11]. *S. marianum* contains a relatively high amount of lipids in the form of triglycerides before processing [4]; the oil content is about 18%–31% and has plenty of unsaturated fatty acids, especially in the form of linoleic acid (42%–54%) and oleic acid (21%–36%). *S. marianum* can therefore also be used in human nutrition [12]. A recent study demonstrated the antatritic and antioxidant effects of milk thistle oil proving the potential to utilize these effects in the food and pharmaceuctical industries [13].

Physical exercise significantly affects physiological and biochemical processes in the body (not only in sport horses). Also, many factors can play a role due to the individuality of horses, such as the type of physical exercise, the duration, the environment, training, rest, and the individual ability to manage stress. The consequences of physical exercise are reflected in the condition and health of individuals, assessed objectively by biochemical changes in the blood before the damage to the organism and metabolic disruption [14,15].

The aim of this study was to evaluate the effect of milk thistle seed cakes (in the form of a granulated mixture with barley) on blood biochemical parameters during feeding (i.e., after 56 days of feeding) and particularly to monitor the differences between the experimental (milk thistle in feed) and control horses after exposure of the monitored horses to heavy physical exercise (combined driving).

## 2. Materials and Methods

### 2.1. Horses and Feeding Model

We used a total of ten horses in the experiment. The horses (one stallion, two geldings, and two mares) aged 10.4 ± 2.0 years were included in the experimental group, and five horses (two stallions, two geldings, and one mare) aged 9.0 ± 2.9 years were placed in the control group. The horses had already been exposed to four months of combined driving training before the beginning of the study. All ten horses were the members of the Czech Warmblood breed bred in the Czech Republic in the experimental monitoring setting. Table 1 describes the characteristics of horses and feeding.

A part of the barley feed dose was replaced by 2 kg of extruded barley granules (all horses received the same granulated mixture) before the start of the experimental monitoring. The horses were divided into groups after 28 days. Up to 2 kg of extruded granules (contained from 80% of barley and 20% of milk thistle) were fed to the experimental group, and the same extruded granules were composed only of barley (up to 100%) were fed to the control group for the next 56 days. The intake of silymarin reached up to 16.6 g/day from the feed dose in the experimental group compared to the group only fed with extruded barley granules (up to 0 g/day of silymarin). The horses were fed twice a day (the horses received half of the feed dose in the morning and the other half in the evening). The silymarin content of the granulated mixture with 20% of milk thistle seed cakes reached up to 8.3 g/kg, so the daily dose of silymarin was up to 16.6 g. 

The horses had free access to salt lick (halite without additives) and water ad libitum. The horses did not show any clinical signs of disease. The horses were always harnessed in pairs, as they were used to, when applying the methodology and when dividing horses into groups. The working pair consisted of one horse from the experimental group and one horse from the control group to ensure the most even physical exercise for all horses in the experimental and control groups. The horses were harnessed in pairs to two four-in-hand teams and one tandem on the day of physical exercise.

### 2.2. Exercise Protocol

Driving horses were harnessed in pairs, and two horses formed each pair (the horses are used to it). The horses in the pair were divided into the experimental (one horse) and control (one horse) groups to ensure the same conditions of physical exercise. A total of three blood samples were taken (second, third, and fourth samples) on the day of monitoring the effect of physical exercise.

Blood sampling was carried out for all ten horses before the start of physical exercise, then at an interval of 15 and 60 min after the end of physical exercise. Physical exercise was represented by normal combined driving training (a total of 2 h). The horses were harnessed to the carriage. Combined driving training such as dressage and obstacle cone driving (not a marathon) took place on different types of surfaces (Table 2). The approximate energy requirement for the work (determined by the calculation from the weight of the horses, carriage, and rolling resistance) was calculated as a percentage of the conservation energy requirement that reached up to 72.4%, so it was hard work [16]. The training took place during summer days. The ambient temperature ranged from 20 °C to 25 °C.

### 2.3. Determination of Silymarin

The content of silymarin was performed by HPLC-UV/VIS instrument (Dionex Ultimate 300, Dionex Softron GmbH, Germering, Germany). Chromatography column Hypersil GOLD Dim (150 × 4.6) (Thermo Fisher Scientific, Waltham, MA, USA) was used for separation by temperature 30 °C. The sample (5 µL) was injected by autosampler. The flow rate was 1 mL/min. Content of the mobile phase was A: 0.1% formic acid, B: 100% methanol. The substances were leaved to infuse in an isocratic elution way (mobile phase A was 65% and mobile phase B was 35%). Detection of separated substances was in motion under circumstances of wavelength 288 nm. 

### 2.4. Laboratory Analysis of Blood Samples

A total of four blood samples were taken. The first blood sample was performed before the start of the milk thistle seed cakes feeding in the experimental group (all horses received the same feed dose before this sampling. Barley feed mixture was received by the horses in the control group during the whole experimental monitoring). The second sample was carried out on all horses after 56 days of milk thistle seed cakes feeding before the start of physical exercise in the experimental group. The third blood sample was performed 15 min after the end of physical exercise, and the last fourth sample was taken 60 min after the end of physical exercise.

Blood samples were taken from the horses from the external jugular vein into a heparin plasma collection tube (Li-Heparin) before the beginning of the experiment (day zero), approximately 2 h after the morning feeding. The third blood sampling was carried out after 56 days after the exposure of the horses to physical exercise. The first sampling was performed before the start of physical exercising. The second and the third sampling were taken after 15 min and 60 min after the end of physical exercise. The blood samples were immediately (in the order of 5 min) placed in a centrifuge and centrifuged at 3200 rpm for 10 min after the sampling. Unturned “whole” blood samples were pipetted separately for gamma glutathione peroxidase determination. Blood and plasma samples were stored in Eppendorf safe-lock tubes (up to 1.5 mL) (Eppendorf, Hamburg, Germany) and frozen (−20 °C) until the laboratory processing. The impact assessment of milk thistle seed cakes on the health of the horses was objectively determined by the statistical analysis of blood biochemical parameters (total protein, Albumin, ALT, AST, ALP, GGT, Bilirubin, Total Cholesterol, HDL, LDL, TAG, BHB, NEFA, Creatine Kinase, Creatinine, Urea, GSH-Px, TAS, Glucose, Cortisol, Ca, Pi) related mainly to liver and kidney health, energy metabolism, and stress. 

Photometric determination of the biochemical blood parameters was performed in a physiological laboratory on the KONELAB T20xt automatic analyzer (Thermo Fisher Scientific, Vantaa, Finland). The kits according to the manufacturer instructions (Biovendor-Laboratory Medicine, Brno-Řečkovice a Mokrá Hora, Czech Republic) were used to determine the following biochemical parameters: the total protein, albumin, bilirubin, urea, creatinine, ALT, AST, ALP, GGT, the total cholesterol, HDL-cholesterol, LDL-cholesterol, TAG, BHB, NEFA (RANDOX, Crumlin, UK), creatine kinase, lactate, glucose, glutathione peroxidase, the total antioxidant capacity (RANDOX, Crumlin, UK), calcium and phosphorus. The immunochemiluminescence determination of cortisol was carried out in a physiological laboratory on an IMMULITE automated analyzer (Siemens Healthineers, Erlangen, Germany) using commercially available cortisol kits (Siemens Healthineers, Erlangen, Germany).

### 2.5. Statistical Analysis

The data from individual groups were obtained from the determined results of the analyzes. Basic statistical parameters (the mean and standard deviation) were performed and calculated in Microsoft Excel. The obtained data were statistically processed in the program STATISTICA.CZ (12.0) using one-way analysis of variance (ANOVA) and Schefe’s test. A value of *p* < 0.05 was considered statistically significant. Tables and charts were processed in Microsoft Office Excel 2018 (products.office.com/en-us/excel).

## 3. Results

A granulated mixture of barley with a 20% proportion of milk thistle seed cakes was fed, and the monitoring of physical exercise effect was performed in the experimental monitoring. No statistically significant difference was found out between the horses in the experimental and control groups. The monitored blood biochemical parameters corresponded to the reference range of values with the exception of slightly increased creatine kinase values ≥3.20 µkat/L and AST ≥4.17 µkat/L. A change in the activities of a number of enzymes occurred during the adaptation of the horses to the training. The AST and creatine kinase activity were about 30% higher in the plasma of the trained horses compared to the untrained horses. The ALP activity was decreased by the training process [17], which corresponded with our results (lower values ALP in all blood samples ≤2.4 µkat/L). The biochemical blood parameters of the horses are given in Table 3 before the beginning of the experimental monitoring.

A second sample was taken 2 h after the morning feeding (as with the first sample) after 56 days of milk thistle seed cakes feeding (400 g/day). Table 4 shows the results of blood plasma analyzes of the horses divided into control and experimental groups (milk thistle granules up to 20%). A statistically significant difference was found out between the AST values of the groups. No statistically significant difference was observed for other biochemical parameters. Higher AST (≥4.17 µkat/L) values (only in the control group), creatine kinase (≥3.20 µkat/L), and lower ALP (≤2.4 µkat/L) values were observed in the horses as in the first sampling.

Significant differences in the blood parameters were observed between the groups in the biochemical values after exposure to physical exercise. Statistically significant differences in AST, NEFA, and Pi values (Table 5) were detected between the groups in the sampling after 15 min after exercise (third sample). The same results as in the second sample (higher creatine kinase values, lower ALP, and higher AST in the control group) were monitored in relation to the reference range of the values.

Statistically significant differences were found between the groups in the values of AST, glucose, cortisol, and phosphorus, which can be seen in the results of the blood parameters taken 60 min after exercise (Table 6). The average NEFA value reached a still higher value in the control group than in the experimental group. Taking into account the reference range of the values, we could conclude that AST was still slightly higher in the control group. Creatine kinase reached a slightly higher value, and ALP was detected at a lower value in all horses. Glucose was found to have a borderline higher value (≥6.4 mmol/L) in the control group. Phosphorus levels approached the lower limit of the reference range (≤1.0 µkat/L) in the control group. 

### 3.1. Biochemical Parameters of Blood Related to Nutritional Status

The individuality of the horses in respect to the changes in blood biochemistry in individual horses (the difference in values between the second and first sampling) was taken into account in assessing the effect of milk thistle seed cakes feeding on the nutritional status. Significant differences (*p* < 0.05) were found out in the statistical processing. Significant differences (*p* < 0.05) were detected between the experimental and control groups changes between the first and second sampling for individuals for the following parameters: total protein, albumin, urea (Figure 1), TAG, the total cholesterol, HDL-cholesterol, LHL-cholesterol (Figure 2) and the enzymatic activities of AST and GGT (Figure 3).

Statistically significant differences (*p* < 0.05) were observed in the total cholesterol, HDL, and LDL cholesterol levels after 56 days of milk thistle seed cakes feeding from the point of view of individual differences in the horses in the experimental and control groups. These values were increased in all horses in the experimental group. Blood cholesterol levels were not changed in the control group. The TAG values were increased in the control group but remained essentially at the same value in the experimental group. The horses of the experimental group received more crude fat (up to 18.4 g) contained in the milk thistle seed cakes. The average fat content in FD (feed dose) reached about 2.5%. Neither differences in the total values of the total cholesterol, HDL and LDL, and TAG were found between the samples nor between the groups.

Furthermore, individual differences (*p* < 0.05) in the levels of the AST and GGT enzymes were found (Figure 3), which were related to liver markers. The AST level within the individual the horses showed that AST decreased in all horses in the experimental group but remained at the same level or slightly increased in the control group. The GGT value decreased slightly in all horses in the experimental group and slightly increased in the control group.

### 3.2. Biochemical Parameters of Blood Related to Physical Exercise

In this part, those biochemical parameters are mentioned, which differed in the abatement values after the exposure of the horse to physical exercise. These results may not associate with milk thistle seed cakes feeding but can be primarily the result of physical exercise. Only significant differences between the groups were considered to be a possible effect of milk thistle seed cakes feeding. The changes in albumin and lactate levels were related to physical exercise.

A graphical representation (Figure 4) of the AST development values in the experimental and control groups. A significant decrease (*p* < 0.05) in AST values after 56 days of milk thistle seed cake feeding in the experimental group could be seen. Physical exercise did not affect the AST values.

The comparison of the NEFA values between the experimental and control groups (Figure 5) showed no difference observed before exercise. A statistically significant difference (*p* < 0.05) was monitored about 15 min after exercise. NEFA values reached about 40% lower in the experimental group compared to the control group.

Figure 6 shows the development of the P_i_ values (PO_4_^3−^). The order of the sampling proved no effect on P_i_ levels, but statistically significant differences in values were observed between the groups after exercise. The values were not changed in the experimental group. A significant decrease was determined in phosphate anion in the control group.

## 4. Discussion

### 4.1. Biochemical Parameters of Blood Related to Nutritional Status

The values of the total protein and urea are related, among other things, to the intake of nitrogenous substances in the feed ration and are indicative of the adequacy of the daily protein dose [18]. The albumin concentration reflects proteosynthetic ability of the liver [19], so the proteosynthesis rate increased in the horses of the experimental group. Silymarin proves the ability to regulate proteosytesis and can increase the albumin levels [20] in the disruption of proteosynthetic liver function (reduced albumin). No horse showed health disorders in our experimental observation. The horses were fed with milk thistle seed cakes and received more crude protein (by 35 g) compared to the previous month when all horses received the same granular mixture. The albumin values and urea did not change in the control group (Table 4; Figure 1). Possibly, the protein contained in milk thistle seed cakes is better used for the horses, and the silymarin effect may also have stimulated the proteosynthetic processes [21]. The FD effect with low (up to 1.5%) and high fat content (up to 11.5%) was compared with plasma concentrations of TAG, NEFA, HDL, and LDL [20]. Ponies with FD of higher fat content had higher HDL concentrations and lower TAG. This trend was confirmed by our results (Table 4; Figure 2), but the differences in values were not significant in the total values. ALP value responds first to an increase in cholestasis [22]. This ALP value was not changed, indicating a total increase in cholesterol levels in the experimental horses, but no pathological effect was seen in cholesterol metabolism. Silymarin has been reported to have the ability to regulate higher cholesterol, triglycerides, and lipoprotein levels [23]. An individual assessment showed a statistically significant difference (Figure 2) after 56 days of milk thistle seed cakes feeding in the experimental group. An increase in HDL values was also observed. Silymarin has the ability to reduce intestinal cholesterol absorption and increase HDL levels [24]. The monitored parameters were detected in the reference range of the values for the horses.

A slight decrease in GGT (Figure 3) could theoretically be related to the antioxidant effects of silymarin because GGT protects the cells against oxidative stress [19]. Many authors report that silymarin feeding reduces higher GGT levels. A significant difference was detected in AST values in the second sample between the experimental and control groups. The assessment of the AST parameter also confirmed this difference (Table 4, Table 5 and Table 6; Figure 3) in the individual horses. A significant reduction in AST values has been observed after milk thistle seed cakes feeding in many studies [25]. The silymarin active substance demonstrably reduced AST values [26]. The feeding of milk thistle seed cakes significantly reduced the AST level in the blood of the horses. A stabilizing effect has also been observed on GGT values using silymarin in liver damage [26,27,28].

### 4.2. Biochemical Parameters of Blood Related to Physical Exercise and Stress

Arfuso et al. [29] found out a positive correlation with an increase in albumin levels in the horses after exercise. The experimental monitoring of the training lasted for almost two hours in summer; therefore, partial dehydration of the organism was the cause of the albumin level increase in the blood of the monitored horses (Figure 4). The intensive training led to significant changes in acid-base balance in the horses, with a significant increase in lactate levels after exercise (Figure 5). A statistically significant difference was found between the values before and after exercise. No difference was observed between the groups of horses. These changes were caused by the release of muscle energy during anaerobic metabolism. A high positive correlation was detected between the plasma lactate value and the intensity of physical exercise in the horses [30]. As can be seen in Figure 6, the physical exercise did not affect the AST level. A significant decrease in the AST value was detected in the experimental group. Apparently, the significant AST decrease was not related to the physical exercise in the experimental group, probably due to the feeding of milk thistle seed cakes in the feed dose.

Cortisol is a major representative of glucocorticoids, commonly referred to as stress hormones. Cortisol released in horses is associated with the length and intensity of exercise [31]. Table 6 shows a significant difference between the groups of the horses in the fourth sample. Significantly lower values were recorded in the experimental group compared to the control one. Previous studies suggested that maximum cortisol concentration was reached earlier in trained horses and these horses had a faster recovery time compared to resting values. On average, maximum cortisol levels were observed approximately 30 min after exercise. A rapid transition is desirable to achieve from catabolic to anabolic state after the training [31], which was confirmed in the experimental group. Thakare et al. [32] described in their study that silymarin reduced corticosterone levels in mice exposed to acute stress. Possibly, the silymarin effect may have increased the resistance to stress in the experimental group and achieved the desired acceleration of the body transition into an anabolic state after physical exercise in sport horses.

Glucose homeostasis is closely related to the endocrine system. Gluconeogenesis is accelerated, and glucose absorption in the tissue is reduced, and insulin sensitivity is lower in response to the leaching of glucocorticoids into the blood. An effect from the sampling order on plasma glucose concentration was not found during statistical data processing. A statistically significant difference was detected between the groups in the fourth sample (Table 6). We suppose that the significant difference in glucose levels between the groups in the fourth sample is related to the significant difference in cortisol levels in the fourth sample taking into account the gluconeogenic effect of cortisol [33]. Neither hypoglycaemia nor hyperglycaemia was observed in the horses during the experimental monitoring.

NEFA are highly energy-rich molecules competing with glucose, especially in aerobic exercise [29], and reflect the lipolysis level. NEFA and TAG are a source of energy in plasma, especially at long-term exercise with low and medium intensity. NEFA mobilization from fat stores begins relatively early to start physical exercising. The oxidative processes begin fully within minutes in horses [31]. The lipolysis level is controlled by catecholamines (adrenaline, noradrenaline, and dopamine) and a decrease in insulin activity [34]. Glucocorticoids (cortisol) also stimulate lipolysis, thus saving glycogen in trained horses [35]. As can be seen in Figure 5, exercise significantly affected the NEFA levels in the blood, which is logical with taking into account the endurance (aerobic) type of exercise. 

The effect of physical exercise on the NEFA level in horses has been addressed by other authors [36,37]. Cortisol stimulates the mobilization of energy substrates by accelerating gluconeogenesis, mobilizing NEFA through lipolysis, and increasing the availability of amino acids by proteolysis at the start of physical exercising (or stress). These processes could theoretically delay the beginning of the total fatigue during physical exercise [31]. Higher NEFA values could appear as beneficial energy sources. Higher NEFA utilization as an energy source leads to a decrease in plasma lactate levels [34]. This process did not occur in the control group. No difference in lactate levels was found out between the groups (the average lactate values were even lower in the experimental group), which means that the higher NEFA level was not used in the control group of the horses. Higher NEFA values may be due to the low ability to use NEFA as an energy source after physical exercise in horses [29]. The increased lipolysis, serving as a major energy source in endurance exercise, does not prevent hepatic and muscle glycogen depletion associated with a steady decrease in blood glucose levels despite the increased NEFA concentrations [35]. In addition, NEFA can slow glycolysis [33], and thus, the ATP recovery from glycolysis can occur. Higher NEFA levels tend to be in the blood at doses with a higher fat proportion in FD, contrary to the experimental group. NEFA utilization was determined to be at a higher level in the experimental group, and NEFA levels were detected lower after exercise, thus, saving the energy. The utilization of the energy sources was found out more efficient in the experimental group discussed below for Pi concentrations. In plasma, NEFA levels could also be related to the higher PUFA (polyunsaturated fatty acid) content within the FD in the experimental group of the horses. Piccione et al. [38] observed lower NEFA values treated with FD with a higher PUFA proportion after exercise in the experimental groups of the horses. The opinion of Assenza et al. [37] is that the lower NEFA levels may be due to either decreased fatty acid mobilization or increased NEFA utilization (as an energy source) after the PUFA supplementation diet in the blood in the experimental group compared to the control one.

Phosphorus is essential in energy metabolism, indispensable in the ATP synthesis and 2,3-diphosphoglycerate (oxygen dissociation from hemoglobin), and can also modulate the activity of several metabolic pathways [39]. Figure 6 shows the P_i_ values (PO_4_^3−^). The sampling order had no effect on P_i_ levels. Statistically significant differences were observed in the values between the groups after exercise. The values were not changed in the experimental group. A significant decrease was detected in the values of phosphate anion in the control group. Doubek et al. [19] concluded that hypophosphatemia could be occurring according to a reference range. But Kraft and Dür [22] stated that P_i_ values were detected at the lower limit of the reference value range. The physiological decrease in P_i_ values is usually due to an increase in insulin levels or P_i_ consumption during glycolysis [19]. Vervuert et al. [40] monitored, among other things, the P_i_ values in the blood of horses before and after physical exercise and observed a significant decrease in the value of all horses after exercise (in the range of 30–120 min). If the lower phosphorus values bordering on hypophosphatemia due to higher insulin levels were detected, no increase would be recorded in glucose concentration in the fourth sample in the control group. Lower P_i_ values were observed probably due to the need for ATP recovery as an energy source, suggesting that the experimental group proved a better use/recovery of the energy sources. No normal decrease was observed in phosphorus values after exercise. Lipolysis was used more efficiently as an energy source (lower NEFA values after exercise in the experimental group), and thus, energy sources were saved from glycolysis.

Muscular work is necessary for the performance of a sport horse, but even small deviations in muscle function can have an effect on performance, coordination, endurance, and work determination. Skeletal muscle damage (e.g., acute rhabdomyolysis) is readily diagnosed by determining AST and CK levels [41]. No pathological conditions occurred in any of the horses after the evaluation of these two biochemical indicators. Slightly increased AST values could be a sign of the training process in the first sample. An increase in the lactate values was observed in all horses after exercise. No difference was found in the CK values catalyzing phosphorylation. 

Silymarin may reduce the impact of acute stress [32,42]. Longer-term effects of glucocorticoids suppress the immune responses and contribute to the development of negative side effects [33], including insulin resistance [43]. The relationship between cortisol and insulin may have been reflected in glucose results in the fourth sample (demonstrably higher glucose levels in the control group than in the experimental group). Silymarin can have the ability to increase insulin sensitivity and reduce elevated insulin levels in addition to its antioxidant properties and may protect the pancreas from toxic effects, among other things [42]. A rapid decrease should occur in plasma cortisol concentrations in healthy, trained horses. Negative health effects can reduce the immunity of sport horses if higher cortisol concentrations persist for a long time after exercise [33]. A more significant decrease was observed in cortisol values one hour after physical exercise in the experimental group, showing again a positive phenomenon in the group of the horses fed with milk thistle seed cakes. Oxidative stress is also associated with physical exercise and stress. A long-term training can increase the markers of the antioxidant system according to the results [34] and thus to increase the resistance to oxidative stress. TAS and GSH-Px analyses served as the markers of the antioxidant capacity. No difference was detected between the groups or the effect of the blood sampling order on these indicators. The changes may not be detected in the antioxidant values immediately after exercise but may appear until 16–24 h after exercise. In addition, the onset of fatigue activates the sympathetic nerve (catecholamine secretion), promoting the NEFA lipomobilization, thereby optimizing the metabolic response to exercise and stress. Among other things, exercise improves energy metabolism at the muscle level, reflected in higher NEFA utilization [36]. If sport horses were fed by milk thistle seed cakes, the NEFA would be more efficient. Also, the adaptation of the horses to training in energy metabolism, stress and regeneration processes would be accelerated and is desirable

## 5. Conclusions

Our results suggest that the feeding of milk thistle seed cakes has a positive effect on horse health and energy metabolism. Lower NEFA values were found after exercise in the horses fed with milk thistle seed cakes. These values indicate a higher utilization of NEFA during exercise, but further research is needed. A faster return of cortisol to the resting values before exercise occurred after the training in the horses fed with milk thistle seed cakes. This conclusion is a phenomenon demanded in sport horses.

## Figures and Tables

**Figure 1 animals-11-00210-f001:**
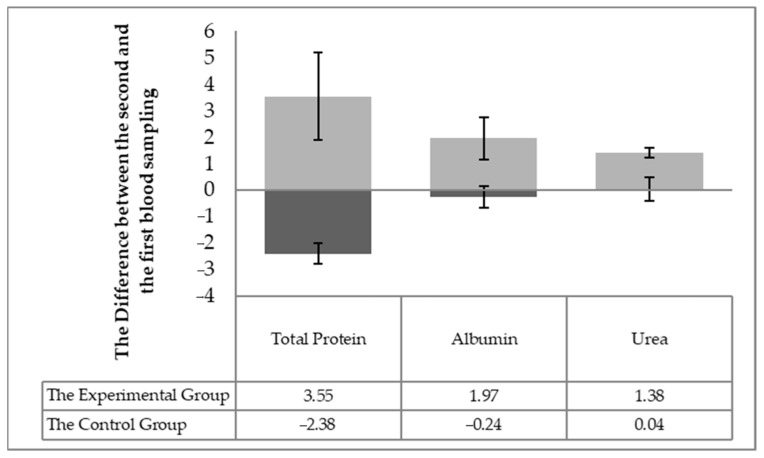
The difference of the monitored parameters (the total protein, albumin, urea) between the first and the second sampling in individual horses.

**Figure 2 animals-11-00210-f002:**
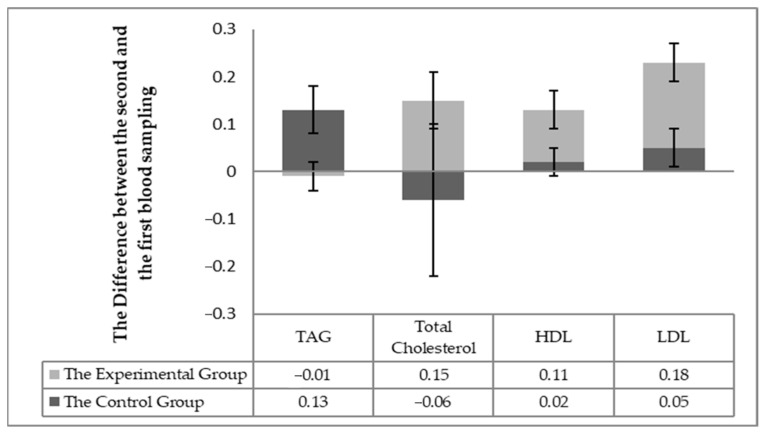
The difference between the monitored parameters (TAG, the total cholesterol, LDL, and HDL cholesterol) between the first and second sampling in individual horses.

**Figure 3 animals-11-00210-f003:**
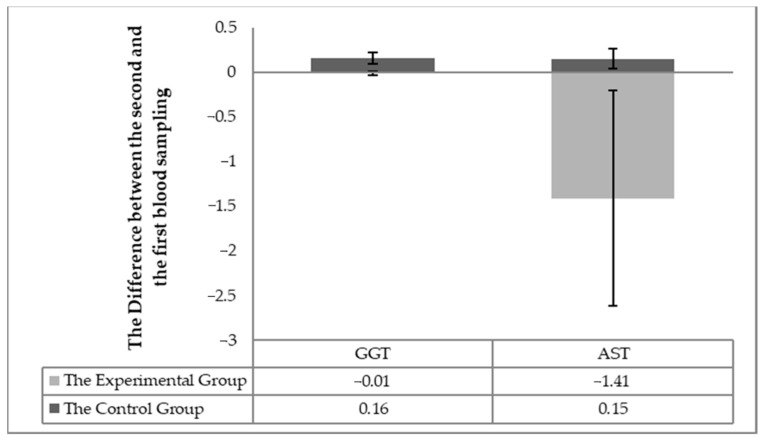
The difference of the monitored parameters (GGT and AST) between the first and the second sampling in the individual horses.

**Figure 4 animals-11-00210-f004:**
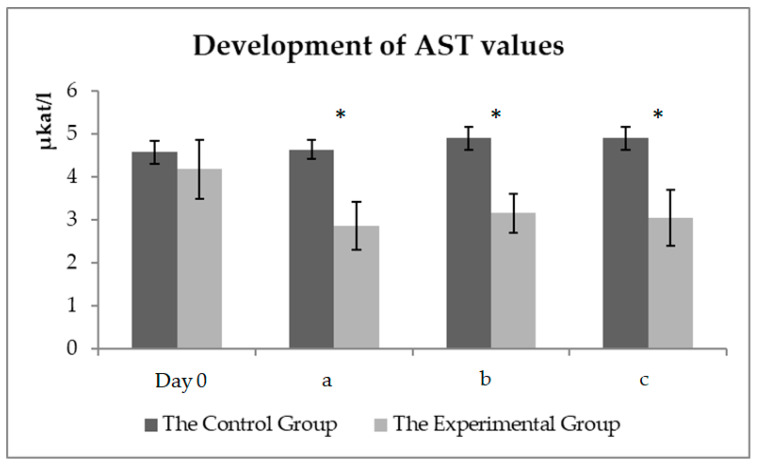
The AST values (µkat/L) in the control and experimental groups. Day zero values before the beginning of the experimental monitoring (day zero)—sample one; (**a**) the values before the start of the training (day 56)—sample two; (**b**) the values about 15 min after the end of the training (day 56)—sample three; (**c**) the values about 60 min after the end of the training (day 56)—sample four. * symbol indicates a significant difference between the control and experimental groups at the level of *p* < 0.05.

**Figure 5 animals-11-00210-f005:**
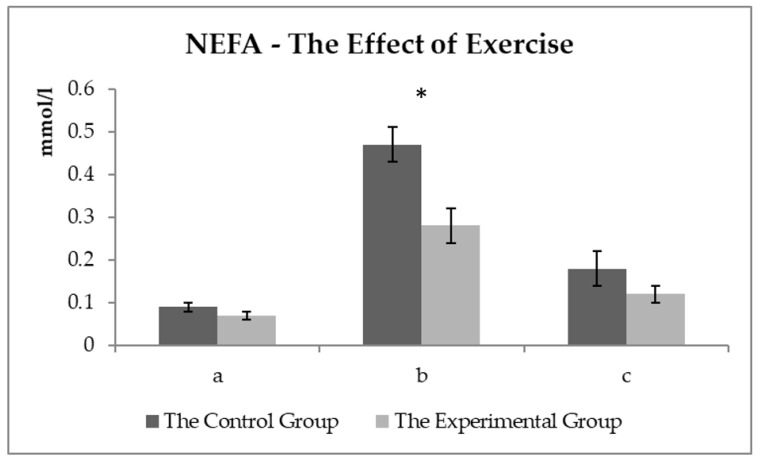
The NEFA values (mmol/L) in the control and experimental groups: (**a**) the values before the start of the training—sample two; (**b**) the values about 15 min after the end of the training—sample three; (**c**) the values about 60 min after the end of the training—sample four. * symbol indicates a significant difference between the control and experimental groups at the level of *p* < 0.05.

**Figure 6 animals-11-00210-f006:**
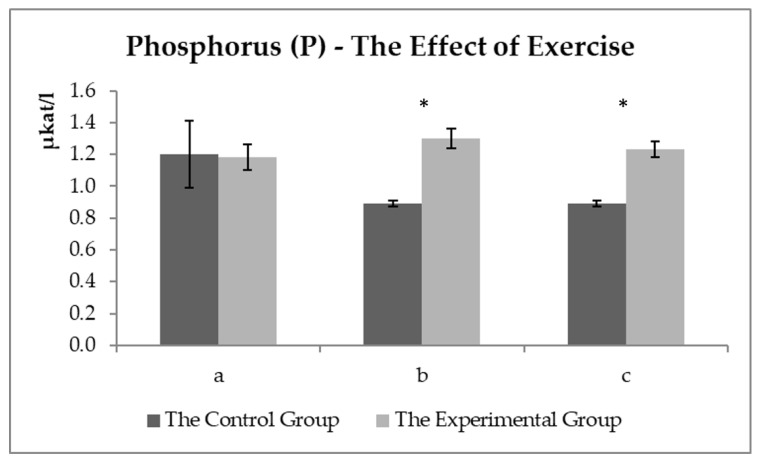
The phosphorus values (µkat/L) in the control and experimental groups: (**a**) the values before the start of the training—sample two; (**b**) the values about 15 min after the end of the training—sample three; (**c**) the values about 60 min after the end of training—sample four. * the symbol indicates a significant difference between the control and experimental groups at the level of *p* < 0.05.

**Table 1 animals-11-00210-t001:** The characteristics of the horses in the experimental monitoring.

Characteristics	Before Experimental Monitoring	Experimental Group	Control Group
Age (years)	9.7 ± 2.9	10.4 ± 2.0	9.0 ± 2.9
Weight (kg)	582 ± 34.2	581 ± 23.2	584 ± 45.7
Body Condition Score (1–5)	3.5	3.5	3.5
**Feeding Regimen (kg/horse)**			
Barley scrap	3	3	3
Barley granules	2	0	2
Barley granules with up to 20% proportion of milk thistle seed cakes/dose of silymarin	0	2/16.6 g	0
Meadow hay	10	10	10
Granulated alfalfa pellet/dryer	1.5	1.5	1.5
**Feed Content (%):**			
Crude fat	2.52	2.64	2.52
Crude protein	12.58	13.11	12.58
Crude fiber	10.75	11.25	10.75

**Table 2 animals-11-00210-t002:** The description of the horse combined driving training (day 56).

Horse Gait	Duration (min)	Velocity (km/h)	Length (m)
**Asphalt Road**			
walk	10 min	5 km/h	833 m
trot	15 min	15 km/h	3750 m
**Grass Surface**			
walk	30 min	5 km/h	2500 m
trot	30 min	15 km/h	7500 m
canter	10 min	30 km/h	1666 m
**Gravel Surface**			
walk	20 min	5 km/h	1666 m

**Table 3 animals-11-00210-t003:** Blood sample one: before the beginning of the experiment on day zero.

Group	Control Group	Experimental Group
*n*	10	10
	average ± SE	average ± SE
total protein (g/L)	69.18 ± 1.90	66.07 ± 1.59
Albumin (g/L)	35.45 ± 0.90	34.45 ± 0.68
ALT (µkat/L)	0.12 ± 0.01	0.13 ± 0.01
AST (µkat/L)	4.57 ± 0.27	4.17 ± 0.68
ALP (µkat/L)	1.62 ± 0.18	1.38 ± 0.15
GGT (µkat/L)	0.24 ± 0.03	0.33 ± 0.04
bilirubin (µmol/L)	26.70 ± 3.38	22.74 ± 2.28
cholesterol (mmol/L)	2.33 ± 0.08	2.33 ± 0.06
HDL-chol. (mmol/L)	1.15 ± 0.05	1.22 ± 0.05
LDL-chol. (mmol/L)	0.96 ± 0.07	0.92 ± 0.04
TAG (mmol/L)	0.20 ± 0.03	0.19 ± 0.03
BHB (mmol/L)	0.23 ± 0.02	0.23 ± 0.01
NEFA (mmol/L)	0.08 ± 0.01	0.08 ± 0.01
creatine kinaze (µkat/L)	3.45 ± 0.82	3.44 ± 0.33
creatinine (µmol/L)	126.54 ± 11.15	132.10 ± 6.58
urea (mmol/L)	4.74 ± 0.34	4.26 ± 0.07
GSH-Px (µkat/L)	607.07 ± 106.94	735.42 ± 36.74
TAS (mmol/L)	1.45 ± 0.08	1.42 ± 0.07
lactate (mmol/L)	0.84 ± 0.01	0.71 ± 0.10
glucose (mmol/L)	5.85 ± 0.28	5.36 ± 0.23
cortizol (µg/100 mL)	4.25 ± 0.67	3.16 ± 0.20
Ca (mmol/L)	2.60 ± 0.06	2.36 ± 0.12
P_i_ (µkat/L)	1.24 ± 0.09	1.04 ± 0.12

SE: mean standard error.

**Table 4 animals-11-00210-t004:** Blood sample two: before the beginning of the experiment on day 56.

Group	Control Group	Experimental Group
*n*	10	10
	average ± SE	average ± SE
the total protein (g/L)	66.81 ± 1.55	70.25 ± 0.89
albumin (g/L)	35.21 ± 0.58	34.45 ± 0.47
ALT (µkat/L)	0.24 ± 0.09	0.15 ± 0.01
AST (µkat/L)	4.63 ± 0.22 ^a^	2.85 ± 0.56 ^b^
ALP (µkat/L)	1.93 ± 0.24	1.88 ± 0.27
GGT (µkat/L)	0.40 ± 0.09	0.29 ± 0.02
bilirubin (µmol/L)	26.81 ± 2.52	26.08 ± 2.60
cholesterol (mmol/L)	2.39 ± 0.07	2.49 ± 0.09
HDL-chol. (mmol/L)	1.18 ± 0.04	1.33 ± 0.08
LDL-chol. (mmol/L)	1.01 ± 0.04	1.10 ± 0.06
TAG (mmol/L)	0.33 ± 0.07	0.17 ± 0.03
BHB (mmol/L)	0.23 ± 0.01	0.26 ± 0.02
NEFA (mmol/L)	0.09 ± 0.01	0.07 ± 0.01
creatin kinaze (µkat/L)	4.09 ± 1.04	3.84 ± 0.28
creatinine (µmol/L)	112.07 ± 9.31	111.86 ± 3.95
urea (mmol/L)	4.78 ± 0.48	5.77 ± 0.28
GSH-Px (µkat/L)	802.84 ± 78.95	764.42 ± 49.37
TAS (mmol/L)	1.46 ± 0.07	1.44 ± 0.02
lactate (mmol/L)	0.67 ± 0.14	0.58 ± 0.06
glukose (mmol/L)	6.20 ± 0.43	6.38 ± 0.22
cortizol (µg/100 mL)	4.53 ± 0.55	5.10 ± 0.49
Ca (mmol/L)	2.53 ± 0.04	2.59 ± 0.04
P_i_ (µkat/L)	1.20 ± 0.21	1.18 ± 0.08

SE: mean standard error. ^a, b^ averages of the same order marked by various letters are significantly different from each other (*p* < 0.05).

**Table 5 animals-11-00210-t005:** Blood sample three: up to 15 min after the end of physical exercise on day 56.

Group	Control Group	Experimental Group
*n*	10	10
	average ± SE	average ± SE
the total protein (g/L)	71.55 ± 1.13	70.59 ± 1.54
albumin (g/L)	38.98 ± 0.90	38.88 ± 0.54
ALT (µkat/L)	0.17 ± 0.03	0.16 ± 0.01
AST (µkat/L)	4.89 ± 0.27 ^a^	3.15 ± 0.45 ^b^
ALP (µkat/L)	2.01 ± 0.20	1.93 ± 0.21
GGT (µkat/L)	0.41 ± 0.10	0.30 ± 0.01
bilirubin (µmol/L)	29.23 ± 1.85	28.31 ± 2.99
cholesterol (mmol/L)	2.49 ± 0.03	2.61 ± 0.09
HDL-chol. (mmol/L)	1.28 ± 0.06	1.38 ± 0.06
LDL-chol. (mmol/L)	1.12 ± 0.02	1.11 ± 0.05
TAG (mmol/L)	0.23 ± 0.01	0.20 ± 0.02
BHB (mmol/L)	0.28 ± 0.02	0.28 ± 0.02
NEFA (mmol/L)	0.47 ± 0.04 ^a^	0.28 ± 0.04 ^b^
creatin kinase (µkat/L)	4.02 ± 0.58	4.38 ± 0.24
creatinine (µmol/L)	128.02 ± 6.45	121.77 ± 5.75
urea (mmol/L)	5.33 ± 0.29	6.12 ± 0.28
GSH-Px (µkat/L)	659.48 ± 18.14	598.84 ± 127.52
TAS (mmol/L)	1.41 ± 0.05	1.44 ± 0.06
lactate (mmol/L)	1.12 ± 0.03	1.09 ± 0.09
glucose (mmol/L)	5.78 ± 0.47	5.95 ± 0.16
cortizol (µg/100 mL)	5.43 ± 0.37	5.58 ± 0.54
Ca (mmol/L)	2.65 ± 0.07	2.62 ± 0.04
P_i_ (µkat/L)	0.89 ± 0.02 ^B^	1.30 ± 0.06 ^A^

SE: mean standard error. ^a, b^ averages of the same order marked by various letters are significantly different from each other (*p* < 0.05). ^A, B^ averages of the same order marked by various letters are significantly different from each other (*p* < 0.01).

**Table 6 animals-11-00210-t006:** Blood sample four: up to 60 min after the end of physical exercise on day 56.

Group	Control Group	Experimental Group
*n*	10	10
	average ± SE	average ± SE
the total protein (g/L)	73.72 ± 1.72	68.99 ± 1.35
albumin (g/L)	39.78 ± 0.93	38.42 ± 0.57
ALT (µkat/L)	0.18 ± 0.04	0.16 ± 0.04
AST (µkat/L)	4.89 ± 0.26 ^a^	3.04 ± 0.64 ^b^
ALP (µkat/L)	2.23 ± 0.25	1.96 ± 0.18
GGT (µkat/L)	0.39 ± 0.11	0.30 ± 0.02
bilirubin (µmol/L)	23.50 ± 2.83	24.39 ± 1.19
cholesterol (mmol/L)	2.53 ± 0.07	2.54 ± 0.09
HDL-chol. (mmol/L)	1.30 ± 0.06	1.33 ± 0.05
LDL-chol. (mmol/L)	1.09 ± 0.01	1.09 ± 0.06
TAG (mmol/L)	0.21 ± 0.02	0.22 ± 0.03
BHB (mmol/L)	0.27 ± 0.03	0.33 ± 0.03
NEFA (mmol/L)	0.18 ± 0.04	0.12 ± 0.02
creatin kinase (µkat/L)	4.16 ± 0.69	4.44 ± 0.27
creatinine (µmol/L)	139.53 ± 9.31	120.67 ± 4.82
urea (mmol/L)	5.24 ± 0.14	5.54 ± 0.28
GSH-Px (µkat/L)	808.84 ± 68.62	780.15 ± 54.42
TAS (mmol/L)	1.45 ± 0.07	1.29 ± 0.04
lactate (mmol/L)	1.01 ± 0.10	0.82 ± 0.06
glukose (mmol/L)	6.63 ± 0.17 ^a^	6.02 ± 0.08 ^b^
cortizol (µg/100 mL)	4.40 ± 0.39 ^a^	3.20 ± 0.32 ^b^
Ca (mmol/L)	2.83 ± 0.04	2.89 ± 0.03
P_i_ (µkat/L)	0.89 ± 0.02 ^B^	1.23 ± 0.05 ^A^

SE: mean standard error. ^a, b^ averages of the same order marked by various letters are significantly different from each other (*p* < 0.05). ^A, B^ averages of the same order marked by various letters are significantly different from each other (*p* < 0.01).

## Data Availability

Data is contained within the article.

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
