# Peer review of "Influence of Milk Thistle (Silybum marianum) Seed Cakes on Biochemical Values of Equine Plasma Subjected to Physical Exertion"

_animals, 2021, doi:10.3390/ani11010210_

Round 1
Reviewer 1 Report
The paper is potentially of interest, but requires re-writing to:
- greatly improve the scientific presentation, in every section of the paper
- improve the English
- focus the discussion
Please refer to the attached and marked up pdf of the manuscript for recommendations.

Author Response
Dear reviewer,
thank you for your recommendations proving a high merit for the improving the manuscript. The explanations for your comments are mentioned below.
Comments in peer-review-9792865.v2.pdf:
Simple Summary, Abstract and introduction have been edited according to your helpful comments.
Line # 82: statements regarding ethical approvals for the study
Blood samples were taken by a veterinarian in accordance with the welfare of the horses in our experimental monitoring.
Line # 89: what other minerals in the salt lick
It was halite without any additives from the Alpine region, NaCl and the composition analysis was not done. It was a common salt lick for animals without any additives available on the market.
Line # 90: Explain the composition of the seed cakes, how they were analyzed for dose of silymarin, how fed to horses
A part of the barley feed dose was replaced by 2 kg of extruded barley granules (all horses received the same granulated mixture) before the start of the experimental monitoring. The horses were divided into groups after 28 days. The experimental group received barley granules with a 20% proportion of milk thistle seed cakes (up to 400 g of seed cakes in the daily dose). The horses also received up to 2 kg of pure barley granules in the control group. The horses were fed twice a day (the horses received half of the feed dose in the morning and the other half in the evening). All horses received a certain daily dose of barley (specifically up to 4 kg). Up to 2 kg of kibbled barley were replaced by 2 kg of barley granules (extruded barley) one month before the start of the experiment. The heat treatment increases digestibility and digestible energy. The horses were divided into 2 groups after this month. The experimental group received only 2 kg of barley granules with a proportion of milk thistle seed cakes (up to 20 %). The daily dose of milk thistle seed cakes reached up to 400 g /day. Milk thistle seed cakes were fed in this form in order to prevent a possible "refusal" of milk thistle seed cakes feeding in their original form by the horses and to ensure that the daily intake is up to 400 g of seed cakes and nothing "remained in the trough". The control group further received up to 2 kg of extruded barley granules throughout the experimental monitoring. The aim was to compare these 2 groups so that up to 400 g of barley was replaced by 400 g of milk thistle seed cakes to detect a difference between the groups of the horses. The silymarin content of the granulated mixture with a 20 % of milk thistle seed cakes reached up to 8.3 g/kg so the daily dose of silymarin was up to 16.6 g. The silymarin content of the milk thistle seed cakes was determined by HPLC-UV/VIS (DionexUltimate 300).
Line # 90: How performed?
It is a type of equestrian sport (combined driving) and the horses must be trained regularly. The training of the experimental monitoring was not different from the usual training to which all horses were exposed before the start of the experimental monitoring. Driving horses are harnessed in pairs so 2 horses form a pair (horses are so used to it). The horses in the pair were divided into experimental (up to 1 horse) and control (up to 1 horse) groups to ensure the same conditions of physical exercise. The horses were harnessed to a carriage. Combined driving training such as dressage and obstacle cone driving (not a marathon) took place on different types of surfaces (Table 2) (https://en.wikipedia.org/wiki/Combined_driving) Phase A2: Dressage and Phase C: Obstacle Cone Driving (not Phase B marathon).
Line # 107: How soon after sampling this occurred?
Blood samples were immediately centrifuged after the sampling (in the order of max. 5 minutes after the sampling) to make the results as relevant as possible.
Line # 192: this statement disagrees with the data presented in tables 5 and 6
The tables 5 and 6 describe the results of the biochemical blood parameters after the exposure of the horses to physical exercise. The figure 2 shows the difference between the second and the first blood samples (without the effect of physical exercise) to monitor the differences in blood values between the individual horses (the individuality of the individual horses). All horses received up to 2 kg of granulated mixture (100% of extruded barley) at the first blood sampling. The horses were divided into the groups at the second sampling. The control group of the horses was fed further with extruded barley (up to 2 kg) and the experimental group of the horses received the granulated mixture (up to 2 kg) with a mixture of milk thistle seed cakes (20% of milk thistle seed cakes and 80% of extruded barley) that means up to 400 g of milk thistle seed cakes every day. The effect of seed cakes feeding and the individual changes were monitored. The first and the second sampling was proved up to 3 hours after the morning feeding when the horses were completely at rest.
Line # 214: incorrect interpretation: read George Brooks recent reviews
The phrase “Lactate levels reflected the rate of anaerobic metabolism during physical exercise” is very simplified. Is it possible to state “Lactate levels reflected the rate stress (for example from anaerobic metabolism) during physical exercise”? Brooks (2020) from the latest findings: Regrettably, misinterpretation of early observations was inappropriate and unfortunate because the events observed were, in fact, strain responses to stresses. In response to cell energy crisis, glycolysis results in ATP production by substrate-level phosphorylation of ADP. When cell work rate is high, lactate produced by driver cells is secreted into the interstitium and circulation from where it can reach a variety of recipient cells such as in heart, liver, kidneys and brain.”. George A. Brooks: Lactate as a fulcrum of metabolism. Redox Biology, 2020, 35, 101454. https://doi.org/10.1016/j.redox.2020.101454
Line # 223: figures 4 and 5 not relevant to the study
Albumin and lactate levels were affected by physical exercise and no difference was found between the groups of the horses - without the effect of milk thistle seed cakes. These graphs are intended to show the effect of physical exercise in the article. The article presents all significant changes in biochemical values therefore Fig. 4 and 5 will be kept in the article.
Line # 241: not relevant to the study
Cortisol levels were further affected by physical exercise. A statistically significant difference was detected between the groups up to 60 minutes after exercise (see Table 6). The symbol of statistical significance * between the groups will be added to the graphs for a better overview.
Line # 253: not relevant to the study
In the case of calcium values, the situation is similar to that of lactate and albumin. The effect of physical exercise is clearly shown.
Line # 264: have not provided any evidence for ergogenic effect.
A possible ergogenic effect can be seen in the possible effect of milk thistle seed cakes on the level of NEFA (possibly higher utilization), a faster decrease in cortisol (effect on the nervous system and metabolism). Of course, further research is needed. We have been working on repeating the experimental monitoring. Further research is needed.
Line # 265: All figures: When there is a significant treatment effect, then show the statistics. Else, delete the figures
The statistical significance between the groups will be added with the symbol *.
Line # 473: Discussion rambles. Need to focus and only deal with putative effects of silymarin, and must be physiological, not statistical.
The discussion has been shortened by half. The effects of milk thistle seed cakes with other contained nutrients are considered. The main influence is assumed for silymarin but a higher residual PUFA content may also play a role.
Reviewer 2 Report
the manuscript entitle "Influence of Milk Thistle (Silybum marianum) Seed 2 Cakes on Biochemical Values of Equine Plasma 3 Depending on Exercise" have the scope to monitor the influence of feeding with milk thistle seed cakes included in the usual feed dose and to detect the effect on biochemical indicators of equine plasma. in the study its is not clear how the authors monitored this effect and what the effect was. Before its pubblication on Animals journal a major revision is needed.
in all manuscript section (Simple summery, abstract and main text) the protocol should be better explain.
Abstract section
line 23 added the control group
lines 27-28 the sentence is not clear
All section should be rewritten
material and method section
line 85 how did youy dived the animals in to two groups?
in the two groups a different feeding regime was administered. how do you justify it?
why the two groups received different percentages of crude protein, crude fat and fiber?
how long the experimental group received the integration?
can you explain the normal combined driving traioning?
the table referiments are not reported in the text
line 126 should be rewritten
the statistical analysis must be remake appling a for repeated measure ANOVA
the results section and discussion should be rewritten on the basis of the new results.
conclusion section is not clear
Author Response
Dear reviewer,
thank you for your recommendations proving a high merit for the improving the manuscript. The explanations for your comments are mentioned below.
the manuscript entitle "Influence of Milk Thistle (Silybum marianum) Seed 2 Cakes on Biochemical Values of Equine Plasma 3 Depending on Exercise" have the scope to monitor the influence of feeding with milk thistle seed cakes included in the usual feed dose and to detect the effect on biochemical indicators of equine plasma. in the study its is not clear how the authors monitored this effect and what the effect was. Before its pubblication on Animals journal a major revision is needed.
in all manuscript section (Simple summery, abstract and main text) the protocol should be better explain.
Abstract section
line 23 added the control group
Up to 10 horses were divided into 2 balanced groups of 5 horses. The horses in the harness form "pairs". Each of these pairs consisted of 1 experimental horse and 1 control horse. Driving horses must be similar, among other things, from an aesthetic point of view - similar weight, body structure, colour…. The assumption was that the horses in pairs perform the same physical activity during physical exercise (together they pull a carriage of a certain weight)
lines 27-28 the sentence is not clear
That was to be meant that all horses received a certain daily dose of barley (specifically up to 4 kg). Up to 2 kg of kibbled barley were replaced by 2 kg of barley granules (extruded barley) one month before the start of the experiment. The heat treatment increases digestibility and digestible energy. The horses were divided into 2 groups after this month. The experimental group received only 2 kg of barley granules with a proportion of milk thistle seed cakes (up to 20 %). The daily dose of milk thistle seed cakes reached up to 400 g /day. Milk thistle seed cakes were fed in this form in order to prevent a possible "refusal" of milk thistle seed cakes feeding in their original form by the horses and to ensure that the daily intake is up to 400 g of seed cakes and nothing "remained in the trough". The control group further received up to 2 kg of extruded barley granules throughout the experimental monitoring.
All section should be rewritten
material and method section
line 85 how did youy dived the animals in to two groups?
Driving horses are harnessed in pairs so up to 2 horses form a pair (the horses are used to it). The horses in the pair were divided into the experimental (up to 1 horse) and control (up to 1 horse) groups to ensure the same conditions of physical exercise.
in the two groups a different feeding regime was administered. how do you justify it?
The aim was to compare these 2 groups that means up to 400 g of barley was replaced by 400 g of milk thistle seed cakes- if there is a difference between the groups of the horses.
why the two groups received different percentages of crude protein, crude fat and fiber?
This was caused by the different content of nutrients in granular mixtures. Barley has a different composition than milk thistle seed cakes.
how long the experimental group received the integration?
The experimental group of horses received in the feed dose up to 2 kg of a granulated mixture with a 20% share of milk thistle seed cakes for 56 days.
can you explain the normal combined driving traioning?
It is a type of equestrian sport (combined driving) and horses must be trained regularly. The training of the experimental monitoring did not differ in any way from the usual trainings to which all the horses were exposed before the beginning of the experimental monitoring.
the table referiments are not reported in the text
It will be rewritten.
line 126 should be rewritten
It will be rewritten.
the statistical analysis must be remake appling a for repeated measure ANOVA
Mathematically, we evaluated the experiment according to the methodologies described by Snedecor and Cochran (1971) as an analysis of variance according to the model (2.2.4.5), where we counted 2 experimental groups, 2 experimental periods, 4 blood sampling periods and 5 repetition.
2 experimental periods --- assumed two physical loads ( four-in-hand team and tandem).
the results section and discussion should be rewritten on the basis of the new results.
conclusion section is not clear
The results and discussion have been rewritten.
Reviewer 3 Report
The title should be changed into "Influence of milk thistle (Silibum marianum) on the biochemical parameters of plasma in horses subjected to physical exertion"
Keywords should be like : silymarin, horse nutrtion, biochemical parametrs, physical exercise
Enzymes activity should be recalculated from µkat/l into IU/l - these units are clearer to international audience
Concentration of Pi should be given in mmol/l not in µkat/l
In the "material and methods" authors should clearly indicate that blood for testing was taken from all horses four times - on day zero before the start of the experiment and three times on day 56 after the use of the nutritional supplement in the experimental group
When citing literature, authors often only use numbers, such as in verse number 303 which begins with "(33) found out......". It should be like that "Arfuso et al (33) found out" Verse 336 "(40) described" it should be "Thakare et al. (40) described"
"Discussion" is too long - should be shortened. Authors cited 60 positions of references - it is to much in original paper. Few references are in Czech language - its titles should be translated into English
In the verse 320 authors wrote that " the diagnosis of muscle damage is essential" They based diagnosis of this disorder on determination of AST activity. it is a pity that the authors in their experiment did not indicate activity of lactate dehydrogenase (LDH) whis is more specific for muscle tissue than AST
Author Response
Dear reviewer,
thank you for your recommendations proving a high merit for the improving the manuscript. The explanations for your comments are mentioned below:
The title should be changed into "Influence of milk thistle (Silibum marianum) on the biochemical parameters of plasma in horses subjected to physical exertion"
The title has been changed.
Keywords should be like : silymarin, horse nutrition, biochemical parametres, physical exercise
Keywords have been changed.
Enzymes activity should be recalculated from µkat/l into IU/l - these units are clearer to international audience. Concentration of Pi should be given in mmol/l not in µkat/l
We decided to choose for the use of "kat" units. The General Conference on Weights and Measures (Conférence Générale des Poids et Mesures) states that unit U is not coherent with SI and the International Federation of Clinical Chemistry and Laboratory Medicine request the unit kat. Conférence Générale des Poids et Mesures decided to adopt the special name katal (symbol kat) for the unit SI mol/s to express the catalytic activity especially in the fields of medicine and biochemistry. https://www.bipm.org/utils/common/pdf/si-brochure/SI-Brochure-9-EN.pdf.
In the "material and methods" authors should clearly indicate that blood for testing was taken from all horses four times - on day zero before the start of the experiment and three times on day 56 after the use of the nutritional supplement in the experimental group
It has been changed
When citing literature, authors often only use numbers, such as in verse number 303 which begins with "(33) found out......". It should be like that "Arfuso et al (33) found out" Verse 336 "(40) described" it should be "Thakare et al. (40) described"
It has been changed
"Discussion" is too long - should be shortened. Authors cited 60 positions of references - it is to much in original paper. Few references are in Czech language - its titles should be translated into English
The discussion has been changed and shortened.
In the verse 320 authors wrote that " the diagnosis of muscle damage is essential" They based diagnosis of this disorder on determination of AST activity. it is a pity that the authors in their experiment did not indicate activity of lactate dehydrogenase (LDH) whis is more specific for muscle tissue than AST
The LDH determination had been planned but unfortunately LDH was overlooked in a long list of analyzes and the laboratory failed to set this parameter.
Round 2
Reviewer 1 Report
Thank you for the improvements to the paper. The concerns identified in the attached 'author response' file still need to be improved and corrected.

Author Response
Vážení recenzenti,
děkuji vám za doporučení, která prokazují vysokou zásluhu na zdokonalení rukopisu. Upravený dokument je uveden níže (dokument v příloze).
